# Redox-responsive peptide-based complex coacervates as delivery vehicles with controlled release of proteinous drugs

Jiahua Wang [1✉], Manzar Abbas [2,3], Yu Huang [1✉], Junyou Wang [4] & Yuehua Li [1✉]

Proteinous drugs are highly promising therapeutics to treat various diseases. However, they suffer from limited circulation times and severe off-target side effects. Inspired by active membraneless organelles capable of dynamic recruitment and releasing of specific proteins, here, we present the design of coacervates as therapeutic protocells, made from small metabolites (anionic molecules) and simple arginine-rich peptides (cationic motif) through liquid-liquid phase separation. These complex coacervates demonstrate that their assembly and disassembly can be regulated by redox chemistry, which helps to control the release of the therapeutic protein. A model proteinous drugs, tissue plasminogen activator (tPA), can rapidly compartmentalize inside the complex coacervates, and the coacervates formed from peptides conjugated with arginine-glycine-aspartic acid (RGD) motif (a fibrinogen-derived peptide sequence), show selective binding to the thrombus site and thus enhance on-target efficacy of tPA. Furthermore, the burst release of tPA can be controlled by the redox-induced dissolution of the coacervates. Our proof-of-principle complex coacervate system provides insights into the sequestration and release of proteinous drugs from advanced drug delivery systems and represents a step toward the construction of synthetic therapeutic protocells for biomedical applications.

[1] Department of Radiology, Shanghai Jiao Tong University School of Medicine Affiliated Sixth People's Hospital, Shanghai 200233, China. [2] Department of Chemistry, Khalifa University of Science and Technology, P.O. Box 127788 Abu Dhabi, UAE. [3] Advanced Materials Chemistry Center (AMCC), Khalifa University of Science and Technology, P.O. Box 127788 Abu Dhabi, UAE. [4] State Key Laboratory of Chemical Engineering and Shanghai Key Laboratory of Multiphase Materials Chemical Engineering, East China University of Science and Technology, Shanghai 200237, China. ✉email: jiahuawang@163.com; yuhuang6y@163.com; liyuehua77@sjtu.edu.cn

Membraneless organelles (MLOs), including stress granules[1], processing bodies[2], and nucleoli[3] constitute a subset of cellular compartments and are involved in various biophysical functions within living cells. These organelles are usually liquid-like, and highly enriched in proteins and nucleic acids[4]. They spontaneously form through a process known as liquid–liquid phase separation (LLPS). The liquid-like nature and absence of a physical barrier on the periphery of organelles allow for the exchange of ions and client molecules between the surrounding solution and the interior core of the organelles. This ultimately enables the concentration of a wide range of micro and macromolecules in the core through partitioning[5]. Some of the MLOs are active droplets, and their formation and deformation can be regulated by chemical reaction cycles[2,6]. Dynamically compartmentalizing proteins into such MLOs plays an important role in regulating biological and signaling processes. For example, upon heat stress, TORC1 (target of rapamycin complex 1) is sequestered by the formation of stress granules (known as protein and RNA MLOs), which leads to the suppression of TORC1 signaling. Conversely, the reactivation of TORC1 signaling is directed through the disassembly of stress granules[7]. Building on this insight, we hypothesized that active coacervates, based on peptides and nucleic acids produced through LLPS, could serve as promising platforms for the controlled delivery of proteinous drugs.

Coacervates for therapeutic use, especially as a drug carrier, have recently garnered increasing interest due to their numerous advantages, including quickly recruiting cargos under aqueous conditions, high loading capacity, high biocompatibility, and greatly prolong retention times of therapeutics[8–11]. Interestingly, peptide-based simple/complex coacervates, with or without membranes can sequester the macromolecules including the enzymes, nucleotides, and small molecules[12–14]. A recent study from Miserez and co-workers has shown that the micrometer-sized peptide coacervate droplets can recruit proteinous drugs. Notably, these liquid droplets are capable of crossing the cell membrane for intracellular delivery of therapeutics[15,16]. Additionally, thermo-responsive elastin-like polypeptide-based coacervate depots have been used to deliver therapeutics with slow-releasing characteristics[10,17]. Mann and co-workers surrounded the coacervate droplets with erythrocyte membranes, demonstrating the potential of nitric oxide-producing coacervate vesicles as synthetic therapeutic cells for in vivo vasodilation[18]. However, the majority of these coacervates are typically composed of long polymers to maximize the intermolecular interactions. Designing and constructing active therapeutic coacervates made from small biomolecules have not been studied.

As we know, metabolism is the fundamental cellular process in which common metabolites such as adenosine triphosphate (ATP) and nicotinamide adenine dinucleotide (NADH) play central roles[19,20]. Recently, notable advancements have been made in creating coacervate models by combining charged metabolites with short peptides, and such minimal compartments could be the ideal systems to study the dynamic behaviors of protocells[21–24]. We thus set out to develop an active coacervate model based on the phase separation of small metabolite NADPH (negatively charged) and arginine-rich peptides (positively charged), to better understand the controllable delivery of therapeutic proteins in a dynamic phase-separated process. Therapeutic protocells developed from these cell-sized coacervate micro-droplets are likely to be beneficial in the context of blood vessels. In this system, we chose tissue plasminogen activator (tPA), the only U.S. Food and Drug Administration–approved thrombolytic drug for acute ischemic stroke[25], as our model therapeutic protein. Our objective was to investigate the impact of active coacervate droplets on tPA delivery efficiency in vitro.

Thrombolytic tPA has been used as a modality to decrease both disability and mortality when administered intravascularly for the treatment of acute ischemic stroke in patients[25,26]. However, direct intravenous infusion of tPA still poses a challenge due to its short circulation time (only 2–6 min) and limited selectivity[27]. These factors can diminish the thrombolytic efficacy and potentially lead to systemic off-target effects, including fatal hemorrhagic events, even when administered within the recommended dosage range[28]. The goal of our study is to elucidate how peptide-based complex coacervate droplets can regulate protein recruitment (sequestration) and release in blood clot models to improve its efficacy.

Herein, we designed active coacervates by using two components where arginine-rich peptides act as positively charged entities and small metabolites act as negatively charged components, and interestingly electrostatic interactions have played a central role in phase separation. We demonstrated that the formation and dissolution of the coacervate can be regulated by two pathways: the metabolic conversion between NADP$^+$/NADPH, and the ligation/shortening of cationic peptides using redox chemistry. These active coacervates can recruit and sequester negatively charged tPA, resulting in a delayed release of the tPA compared to a free solution in a halo blood clot model. To enhance the delivery efficiency further, we have introduced modifications to the coacervate-forming peptide by incorporating targeting ligands. For instance, when coacervates are formed with fibrinogen-derivative RGD sequence-terminated peptides, they mimic the "bridging effect" between fibrinogen and integrins GPIIb/IIIa ($\alpha_{IIb}\beta_3$) on the surface of activated platelets at the thrombus site[29,30]. As a result, our coacervate-based delivery vehicles have exhibited efficient targeting of thrombus sites. We have also shown that the rapid tPA release can be triggered by redox reactions, including oxidation of the NADPH or the reduction of the disulfide bonds in the longer peptides. These reactions lead to the dissolution of the coacervates, triggering the subsequent release of the tPA. Taken together, our designer active coacervates offer a promising approach to developing therapeutic protocells made of small biomolecules, enabling controlled delivery of proteinous drugs.

## Results and discussion

**Metabolite NADPH forms stable coacervates with arginine-rich oligopeptides.** Small multivalent anionic metabolites and cationic peptides can undergo spontaneous phase separation, forming liquid droplets known as complex coacervates under aqueous conditions[21]. These small phosphate molecules, such as ATP and NADPH, play pivotal roles in living systems, serving as essential biochemical energy sources for cells[31]. We were intrigued by the possibility that metabolism within these liquid droplets could influence the trafficking of protein cargos, potentially contributing to the functioning of biomimetic coacervate protocells. To explore this hypothesis, we developed a model coacervate system enriched with redox-active metabolite NADPH. This coacervate system showed increased stability compared to ATP/peptide coacervates (Supplementary Figs. 1 and 2). Moreover, it facilitated redox-induced assembly and disassembly of the droplets (Fig. 1). NADPH is a tetravalent anion, and the NADPH/NADP$^+$ redox couples are the major determinants of the cellular redox state[32]. To better understand the effect of multivalency and sequence of peptides on the LLPS with NADPH/NADP$^+$, we determined the droplet-forming tendency of NADPH/NADP$^+$ with simple arginine-rich peptides in a range of lengths and charge block abilities. The simple arginine-rich peptides were chosen as the model cations because they are known to form more stable coacervates with small molecules like adenosine monophosphate

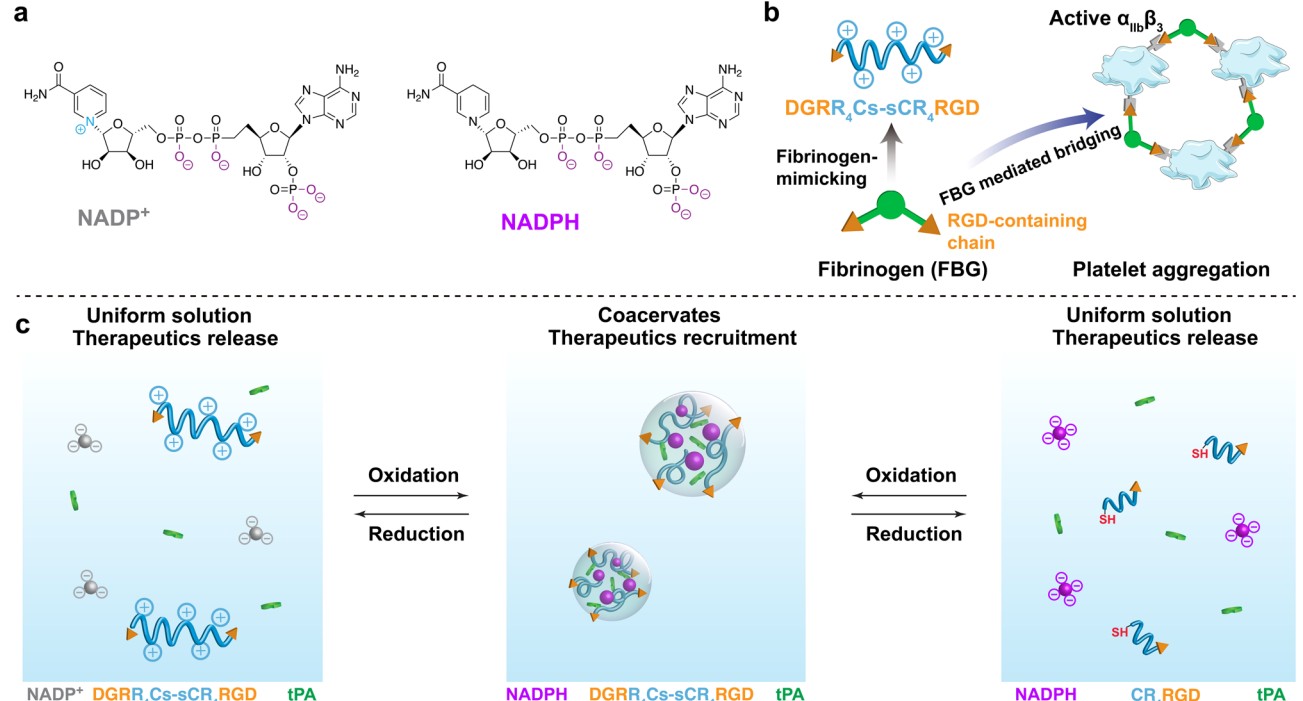

**Fig. 1 Schematic illustration of the redox-active coacervate droplets that mimic the function and structure of fibrinogen as delivery vehicles.**
**a** Chemical structure of NADP$^+$ and NADPH. **b** Design of a fibrinogen-mimicking peptide DGRR$_4$Cs–sCR$_4$RGD as a building block of coacervate droplets, and the mechanism of platelet aggregation through "bridging effect" between fibrinogen and activated platelets. **c** The reversible formation/dissolution of coacervates, the recruitment of therapeutic proteins, and release can be controlled by either the transition of NADPH/NADP$^+$ or the length of the peptides.

(AMP) and adenosine diphosphate (ADP), compared with frequently used oligolysine in coacervate-based studies[33,34].

We initially examined the ability of various short arginine-rich peptide sequences (R$_3$C, R$_4$C, R$_5$C, DGR$_5$C, R$_{10}$, or R$_{30}$) to form complex coacervates with NADPH, NADP$^+$, or NADH in salt-free solution. Through initial screening of phase separation, we found that the shortest peptide sequence comprising four arginine residues (R$_4$C) could form coacervates with NADPH, and R$_{30}$ formed coacervates with NADH, while interestingly no phase separation was observed for NADP$^+$ with all aforementioned arginine-rich peptides (Fig. 2e). The differences in phase separation observed among the NADPH (−4), NADP$^+$ (−3), and NADH (−2) can be attributed to their differing overall valency. In particular, the formation of coacervate droplets through phase separation occurred rapidly and spontaneously with NADPH due to its higher multivalency. Notably, the K$_{20}$ is the shortest oligolysine that is capable of forming complex coacervates with NADPH (Supplementary Fig. 3). This difference in phase separation behavior can be attributed to the higher p$K_a$ of the basic residue in arginine when compared to lysine. The difference in p$K_a$ values suggests that arginine may form more stable cationic–π interactions with NADPH[34–37]. In addition to this, the charge interspacing within the peptide sequence may also influence the properties of NADPH/peptide complex coacervates. For instance, when we substituted R$_5$C with (RG)$_5$C and then with (RGG)$_5$C, we observed a surprising decrease in the stability of the coacervate phase (Supplementary Fig. 3). Higher charge interspacing in (RGG)$_5$C peptide prevents the formation of complex coacervates with NADPH in a salt-free solution. This observation suggests that an increased charge density in short arginine-rich peptides could lead to stronger π–π interactions[37], either between arginine residues or between arginine residues and NADPH dinucleotide, ultimately leading to the formation of more stable coacervates.

As it is evident from the literature, the associative phase separation is primarily driven by charge–charge interactions. The attraction forces between oppositely charged molecules are weakened by charge screening when salt is added[38]. We further investigated the stability of complex coacervates under physiologically relevant conditions (phosphate-buffered saline (PBS), pH 7.4, ionic strength of 0.17 M). Combinations of NADPH/(R)$_{n≥5}$ exhibit greater robustness in forming coacervates in a PBS buffer (Supplementary Fig. 4). This quality renders them as stable coacervate models for further investigation into protein drug trafficking under physiologically relevant conditions. Microscopic investigation reveals the formation of NADPH/R$_{10}$ droplets in PBS buffer, with a typical size range of 5–15 µm (Fig. 2a). The fluorescence properties of nicotinamide moiety of NADPH can help to label the coacervate droplets[39]. As shown in Fig. 2b, the intense blue fluorescence within the coacervates indicates that NADPH is highly concentrated in coacervates, with a partition coefficient of 390 (Fig. 2c, d).

**Redox-responsive behavior of coacervate droplets**. We exploited the redox chemistry of droplets according to the small library of coacervate combinations presented in Fig. 2e. It was interesting to explore how metabolite conversion could trigger phase separation, and conversely, whether metabolic reactions within coacervates could impact the stability of coacervates, potentially leading to dissolution. To assess the potential of a metabolic reaction to induce active coacervate formation, we conducted an enzymatic conversion of NADP$^+$ to NADPH, as illustrated in Fig. 3a. To carry out this investigation, we prepared a solution in PBS buffer containing 4 mM NADP$^+$, 20 mM R$_{10}$, 20 mM glucose-6-phosphate (G6P), and 40-unit mL$^{-1}$ glucose-6-phosphate dehydrogenase (G6PD). As expected, G6PD can catalyze the generation of NADH from NADP$^+$ in the presence of G6P. Within 2 min, this transformation changed a previously

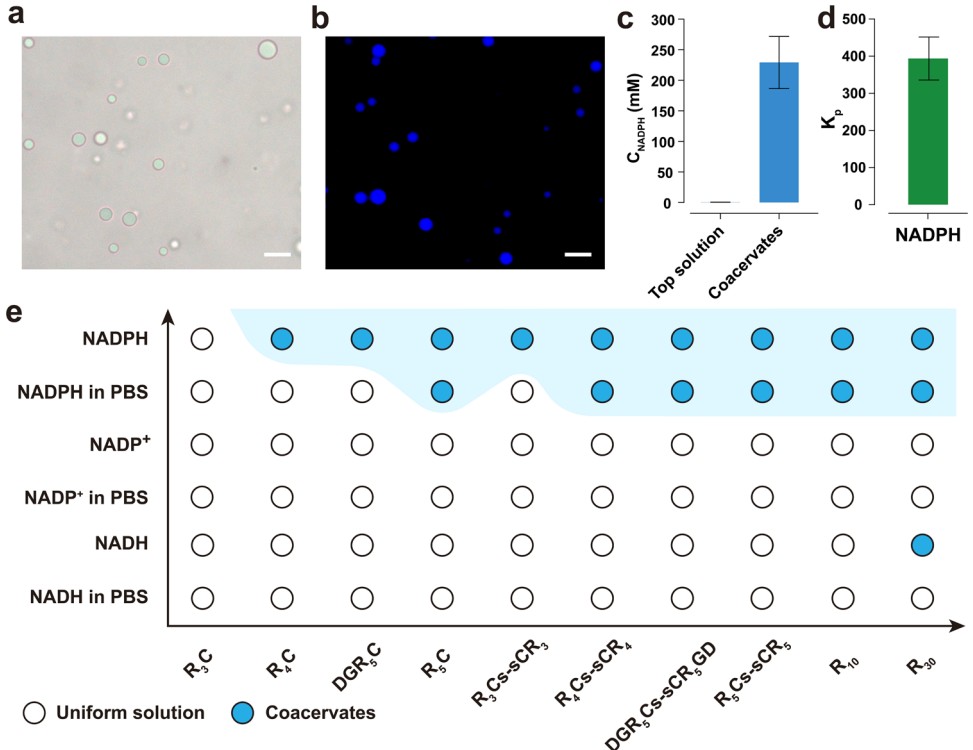

**Fig. 2 NADPH forms stable coacervate droplets with simple arginine-rich oligopeptides. a, b** Brightfield and fluorescence microscope images of NADPH/$R_{10}$ droplets prepared at 4 mM NADPH and 20 mM $R_{10}$ (monomer basis) in PBS buffer. Scale bar, 20 μm. **c** NADPH concentration in the supernatant and coacervate phase. **d** The partition coefficient of NADPH was then calculated using the data obtained in (**c**) (see Supplementary Methods and Supplementary Data 1). **e** Coacervation results obtained by combining pairs of peptides and metabolites. Metabolites (NADPH, NADP$^+$, NADH) 4 mM were mixed with arginine-rich peptide at 20 mM (arginine monomer basis). For a pair of metabolites and $R_5C$ in PBS buffer, the concentration of metabolite was fixed at 8 mM. $R_5C$ forms clear liquid coacervate droplets with NADPH in PBS but requires higher concentrations. Symbols indicate the observation of uniform solutions (white circles) or coacervates (blue circles). Images are obtained over analysis of at least three independent trials. All error bars represent standard deviation ($n = 3$).

uniform solution into a dispersion of NADPH-condensed coacervates (Fig. 3b and Supplementary Data 2). Our study also aimed to investigate whether the NADPH consumption could lead to the dissolution of the coacervate phase, thereby forming a single-phase solution. In contrast, lactic dehydrogenase (LDH) was employed to dissolve the coacervates by converting NADPH into NADP$^+$ using pyruvate as substrate. Notably, NADPH/$R_{10}$ coacervates displayed reversibility, as they could be formed and dissolved within the same solution through the addition of specific reagents (Fig. 3b).

Reduced glutathione (GSH) plays a critical role in chemically detoxifying reactive oxygen species such as hydrogen peroxide ($H_2O_2$), transforming into oxidized glutathione (GSSH), which loses its protective properties[40]. This process can be reversed through the catalytic action of glutathione reductase, using NADPH as a cofactor[41]. Therefore, we asked if the formation and dissolution of NADPH-based coacervate could also be chemically controlled by $H_2O_2$ and GSH. To test this hypothesis, we designed the arginine-rich peptides with a cysteine residue ($R_4C$, $R_5C$, or $DGR_5C$) to investigate the redox properties of the NADPH-based coacervate system. Upon oxidation of the free thiols using $H_2O_2$, the fragments of arginine peptides with thiols are converted into longer arginine-rich peptides with higher multivalency. This transformation results in a transition from a transparent solution to a turbid dispersion of coacervate droplets (Figs. 2e and 3d). This change in phase behavior is entirely reversible through disulfide bond reduction by GSH (Fig. 3d and Supplementary Data 3). It is worth noting that the $H_2O_2$ alone cannot oxidize NADPH, as the activation energy barrier is too

high in the absence of a catalyst (Supplementary Fig. 5)[42]. Subsequent addition of $H_2O_2$ or starting material (NADPH or $CR_5GD$) does not result in the recovery of coacervates. This is mainly attributed to the formation of byproduct, GSHs–s$R_5GD$. Notably, the utilization of tris(2-carboxyethyl)phosphine (TCEP) as a reducing agent enables the restoration of this redox cycle, making it repeatable once again (Supplementary Fig. 6).

**Partitioning of tPA into coacervate droplets.** Macromolecules such as proteins, enzymes, and nucleotides can be spontaneously compartmentalized into bimolecular condensates (coacervates) formed through LLPS. This phenomenon is increasingly recognized as a fundamental principle governing the regulation of cellular processes[43,44]. One of the most important characteristics of coacervates is their ability to uptake and concentrate a wide variety of guest molecules[45]. Consequently, we assessed the ability of the NADPH/peptide coacervates to concentrate a therapeutic protein drug tPA (Fig. 4). tPA was found to concentrate within the coacervates, with an apparent partition coefficient of 55 (Fig. 4e, f). To visualize the internalization and sequestration of tPA within coacervates, we labeled tPA with fluorescein isothiocyanate (FITC). The fluorescence intensity of FITC was significantly higher inside the coacervate droplets (Fig. 4c, d), providing clear evidence of the successful tPA encapsulation. Interestingly, an excess of tPA can decrease coacervate stability (Supplementary Fig. 7). This phenomenon is primarily attributed to the competition between the negatively charged tPA and peptides within the coacervates. The disassembly of coacervates was exploited for the release of the loaded tPA. For example,

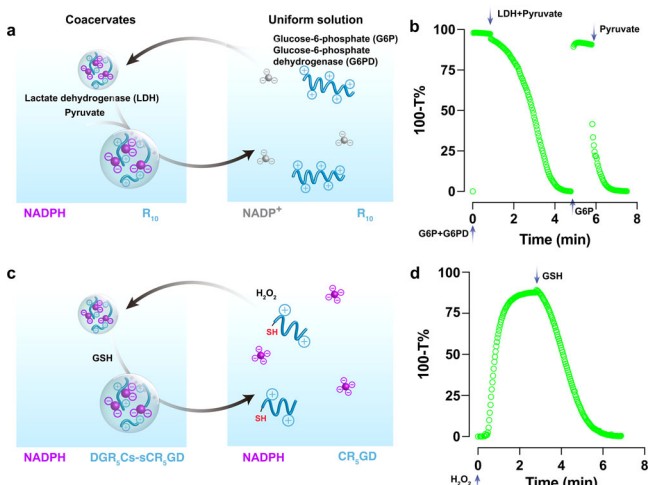

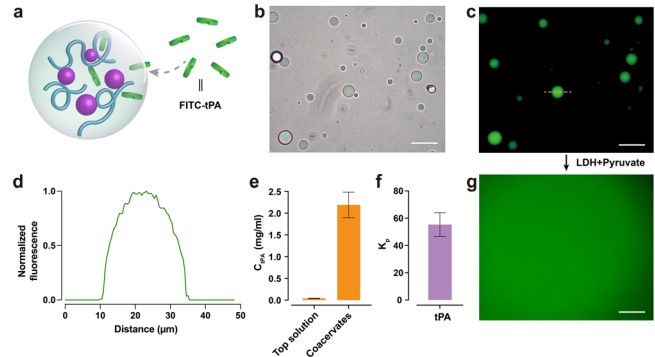

**Fig. 3 Redox-responsive behavior of coacervate droplets. a** Schematic illustration of the enzymatic reaction network underlying the reversible formation and dissolution of NADPH/R₁₀ coacervate droplets. **b** The alternate addition of LDH and pyruvate mixture or G6P and G6PD mixture demonstrated a dynamic cycle of condensation and dissolution of NADPH/R$_{10}$ coacervate droplets. These mixtures were prepared in PBS buffer, at the concentration of NADP$^+$ (4 mM), R$_{10}$ (20 mM) (monomers basis), pyruvate (20 mM), LDH (40 U mL$^{-1}$), G6P (20 mM), G6PD (40 U mL$^{-1}$). **c** Schematic illustration of the redox reaction network underlying the formation and dissolution of NADPH/DGR₅Cs-sCR₅GD coacervate droplets. **d** The alternate addition of H$_2$O$_2$ and GSH shows condensation and dissolution of NADPH/DGR₅Cs-sCR₅GD coacervate droplets. These mixtures were prepared in PBS buffer, at the concentration of NADP$^+$ (4 mM), R$_{10}$ (20 mM, monomers basis), H$_2$O$_2$ (2.5 mM), and GSH (2.5 mM).

adding the LDH–pyruvate mixture into the tPA-loaded coacervates solution showed a release of tPA into the surrounding aqueous phase (Fig. 4g and Supplementary Fig. 8).

**In vitro tPA delivery**. To take one step further, we examined how the release of encapsulated protein drug tPA from complex coacervates, in conjunction with redox chemistry, may influence blood clotting in thrombosis. Thus, we conducted a halo blood clot assay, as illustrated in Fig. 5a, to assess the impact of redox-active coacervates on the controlled release of tPA. In this assay, halo-shaped blood clots with an empty center in 96-well plates were treated with different tPA-containing formulations. We monitored the clot dissolution by measuring absorbance at 510 nm, which indicates the release of red blood cells during thrombolysis[46,47]. As shown in Fig. 5c, free tPA completely dissolved the halo-shaped blood clots after 1 h of treatment at 37 °C. The lysis of blood clots induced by tPA-NADPH/R₅Cs-sCR₅ (tPA-NADPH/Rpep) coacervates exhibited an initial delay of approximately 24 min compared to free tPA. This delay might be attributed to the isolation of tPA within coacervates, resulting in a gradual release of tPA from the coacervate interior[48].

It was both logical and interesting to investigate how the peptide sequence, a component of coacervate formation, could facilitate thrombolytic activity. The fibrinogen-derivative RGD motifs have been reported for their specific binding ability to stimulate $\alpha_{IIb}\beta_3$ integrins abundantly expressed on the surface of activated platelets at the thrombus site[30,49]. To determine whether incorporating the RGD derivative of coacervate-forming peptides could improve the selectivity of coacervates in binding to activated platelets, we conducted microscopic experiments to assess the coacervate-platelets binding capacity. As shown in Fig. 5e, RGD motifs-linked peptide-based NADPH/

**Fig. 4 Partitioning of tPA into coacervate droplets. a** Schematic illustration of associative liquid-liquid phase separation of NADPH and R₁₀ to produce tPA-encapsulated coacervate protocells. tPA, tissue plasminogen activator. **b**, **c** Bright-field image of NADPH/R₁₀ coacervate droplets encapsulating FITC-labeled tPA, and corresponding fluorescence image of NADPH/R₁₀ coacervate droplets. Localization of fluorescence within the droplets confirms tPA encapsulation. Scale bar, 50 μm. **d** Corresponding intensity profiles (along the yellow dashed line) of fluorescence images of FITC-tPA fluorescence (green). Fluorescence intensities were normalized by the maximum intensity. **e** tPA concentration in the supernatant and coacervate phase. **f** The partition coefficient of tPA was then calculated using the data obtained in (**e**) (see Supplementary Methods and Supplementary Data 4). **g** Release of FITC-tPA from coacervates through LDH–pyruvate mixture addition. All error bars represent standard deviation (n = 3).

DGR₅Cs-sCR₅GD (NADPH/Rpep-RGD) coacervates exhibited a notable binding with activated platelets (Supplementary Fig. 9). In contrast, NADPH/Rpep-RGD coacervates caused negligible binding with resting platelets (Fig. 5d). Likewise, the NADPH/Rpep-RGD coacervates demonstrated enhanced binding to Hela cells through the specific recognition of $\alpha_V\beta_3$ integrins on the of tumor cell membrane[50] (see Supplementary Fig. 10). Interestingly, when coacervates incubated with platelets, their coalescence was weakened, these might be attributed to the presence of fragments from erythrocyte or platelet membranes coating the surface of coacervate droplets (Supplementary Figs. 11 and 12)[18,51–53].

To broaden the application of this finding, we evaluated the thrombolytic activity of tPA-loaded NADPH/Rpep-RGD coacervates as a delivery system in a blood clot model. As shown in Fig. 5c, the thrombolytic activity of tPA-containing NADPH/Rpep-RGD coacervates is higher than tPA-NADPH/Rpep coacervates. This observation undercores the potential of tPA-NADPH/Rpep-RGD coacervates with the RGD motif to efficiently facilitate clot lysis by targeting the thrombus site. Finally, we explored whether the isolated tPA within the coacervate interior could be rapidly released through coacervate dissolution. Given the redox-active nature of these coacervates, we added an LDH–pyruvate mixture to oxidize NADPH into NADP$^+$, leading to the dissolution of coacervates. As expected, the dissolution of tPA-loaded coacervates induced a complete release of tPA at the thrombus site, resulting in a fast clot lysis upon the addition of LDH–pyruvate mixture. This study highlights how modifying a peptide sequence with proven targeting motifs can be a crucial parameter in designing complex coacervate systems for controlling the release of protein drugs. Moreover, in certain cases, this approach allows for precise control over the desired burst release by redox chemistry.

## Conclusions

We have developed an innovative type of active coacervate system, which forms *via* the phase separation of anionic metabolite

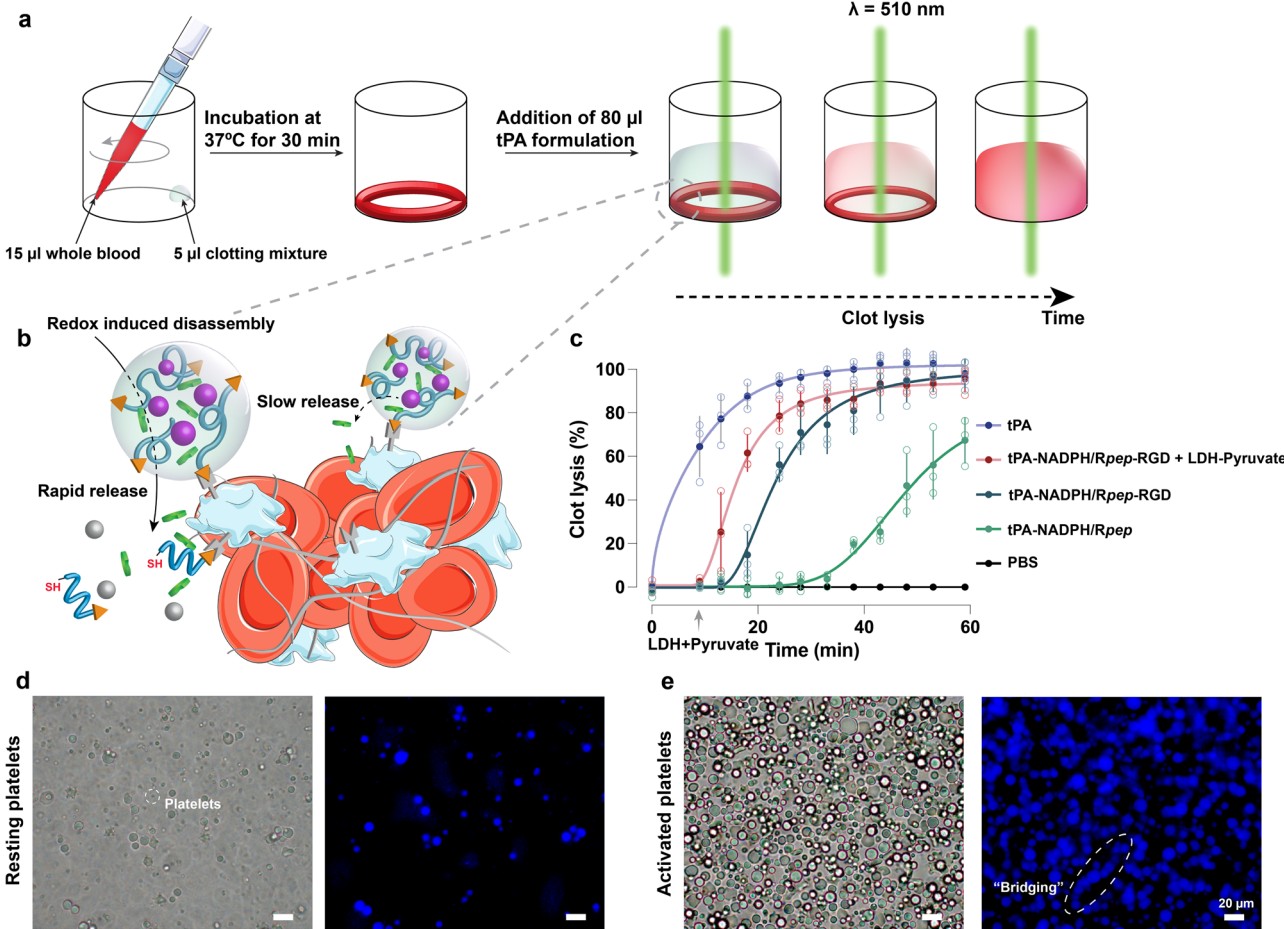

**Fig. 5 The peptide sequence and redox-responsive behavior of coacervate droplets determine the efficiency of tPA delivery. a** Schematic illustration of the clot assay protocol. Blood was pipetted along the wall of wells in a 96-well plate, and the formation of clots was accomplished by adding a clotting mixture and incubating at 37 °C for 30 min. **b** A schematic illustration of the thrombolytic activity of tPA-NADPH/R*pep*-RGD coacervates shows that they can be rapidly activated through the redox-induced dissolution of the coacervates. **c** Time-dependent clot lysis in the halo blood clot model after treatment with PBS buffer (pH 7.4) only, tPA-NADPH/R*pep*-RGD coacervates, tPA-NADPH/R*pep*-RGD coacervates (dissolution of coacervates by addition of LDH–pyruvate mixture at 9 min), tPA-NADPH/R*pep* coacervates, and free tPA (see Supplementary Methods and Supplementary Data 5). **d** Bright-field image and the corresponding fluorescence image of resting platelets incubated with NADPH/R*pep*-RGD coacervates. **e** Bright-field image and the corresponding fluorescence image of activated platelets incubated with the NADPH/R*pep*-RGD coacervates. Scale bar, 20 μm. All error bars represent standard deviation ($n = 3$).

NADPH with short arginine-rich peptides. The generation and dissolution of these coacervates can be precisely controlled through two pathways: metabolic conversion between NADP$^+$/NADPH couples and ligation/cleavage of the cationic peptides using redox chemistry. Since a wide range of biomacromolecules (such as proteins, peptides, and RNAs) can be efficiently recruited by coacervates. The versatility of these redox-responsive coacervates positions them as an exceptionally promising model compartment for in-depth studies on the recruitment and precisely regulated release of biomacromolecules, a process tightly controlled by metabolic reactions. We harnessed these coacervates as delivery vehicles for protein-based therapeutic, exemplified by the model thrombolytic drug tPA. These coacervate droplets efficiently served as tPA reservoirs, enabling controlled release through the chemical system design. Targeted delivery was achieved by incorporating an RGD sequence into coacervate-forming peptides, mimicking the structure and function of fibrinogen. Our NADPH/R*pep*-RGD coacervate-based delivery vehicles exhibited specific binding to the abundant activated platelets at the thrombus sites. The efficient release of tPA could then be triggered by the redox-induced dissolution of the coacervates. In short, this biomimetic coacervate system holds promise as a model for constructing smart therapeutic protocells composed simply of biomolecules.

## Methods
A full description of materials and methods used in this work is given in the Supplementary Information.

**Reporting summary**. Further information on research design is available in the Nature Portfolio Reporting Summary linked to this article.

## Data availability
The data supporting the findings of this study are available within the paper and its Supplementary Information. Numeric data underlying all plots in the main publication are available as Supplementary Data 1–5. Source data are provided with this paper.

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

## Acknowledgements

This work was supported financially by the Fundamental Research Funds for the Shanghai Sixth People's Hospital (X-2430 to J.W.; X-2362 and ynqn202111 to Y.H.). This work was also supported by the Innovative Research Team of High-Level Local Universities in Shanghai. The authors also acknowledge the sponsorship from the National Natural Science Foundation of China (22205137 to Y.H.) and the Shanghai Pujiang Program (21PJ1411700 to Y.H.). M.A. acknowledges the financial support (Project No. 8474000462) from Khalifa University of Science and Technology, UAE.

## Author contributions

J.H.W., Y.H. and Y.H.L. conceived the project. Y.H.L. guided and supervised the project. J.H.W. and Y.H. designed the in vitro experiments. J.H.W. and J.Y.W. carried out the synthesis and characterization of coacervates. M.A. assisted in analyzing the data. J.H.W. and M.A. wrote the paper, with input and revisions from all authors.

## Competing interests

The authors declare no competing interests.
