## [Peer Review File · Communications Chemistry]

This manuscript has been previously reviewed at another Nature Portfolio journal. This document only contains reviewer comments and rebuttal letters for versions considered at Communications Chemistry.

REVIEWERS' COMMENTS:

Reviewer #1 (Remarks to the Author):

The authors have addressed most of the comments raised and the paper is now more sound and clearer to read.

I have a few comments that I believe the authors could look at before final acceptance:

- Although the language has been much improved, there remains quite a lot of typos (singular vs. plural used in wrong locations, missing articles in some sentences, etc...). Please have another careful read by an external reader to correct these grammatical mistakes.
- As an example, the second sentence of the abstract is missing an active verb and the next sentence is unclear due to the use of dashes. I would recommend using brackets to clearly explain what are the building blocks of the coacervates.
- The bridging effect compared in Figs. 5d and e and Supp. Fig. 11 is not very obvious. How do the authors define it? It would help to explain in the figure legend or to point out on the figure where is this bridging effect.
- For the sake of clarity, it would also help to use shorter names/acronyms to define the different peptides used, because the naming used at the moment (e.g. tPA-NADH-DGR5Cs-CsR5GD) is quite tedious to read, especially in the Figures' labels and in the captions. Perhaps a Table with the peptide sequence and a short name for each peptide could be included to facilitate the reading.

Reviewer #1:

The authors have addressed most of the comments raised and the paper is now more sound and clearer to read.

I have a few comments that I believe the authors could look at before final acceptance:

► We thank the reviewer for the carefully assessment on the manuscript in general. Following your suggestion, we have carefully polished the text and figures in the revised manuscript to enhance its clarity for the readers.

- Although the language has been much improved, there remains quite a lot of typos (singular vs. plural used in wrong locations, missing articles in some sentences, etc...). Please have another careful read by an external reader to correct these grammatical mistakes.

► We thank the reviewer for the careful proofreading. We went through the text and corrected the grammatical errors; the changes are highlighted in the revised Ms.

- As an example, the second sentence of the abstract is missing an active verb and the next sentence is unclear due to the use of dashes. I would recommend using brackets to clearly explain what are the building blocks of the coacervates.

► As suggested, we went through the text and corrected the grammatical errors.

- The bridging effect compared in Figs. 5d and e and Supp. Fig. 11 is not very obvious. How do the authors define it? It would help to explain in the figure legend or to point out on the figure where is this bridging effect.

► We agree with this point, and we have updated the figures with highlighted bridging effect in the figures.

- For the sake of clarity, it would also help to used shorter names/acronyms to define the different peptides used, because the naming used at the moment (e.g. tPA-NADH-DGR5Cs-CsR5GD) is quite tedious to read, especially in the Figures' labels and in the captions. Perhaps a Table with the peptide sequence and a short name for each peptide could be included to facilitate the reading.

► We thank the reviewer for this comment. As recommended, we have clearly defined the coacervates employed in our in vitro clot experiments (Figure 5). Acronyms include tPA-

NADPH/*Rpep*-RGD (tPA-NADPH/DGR5Cs-sCR5GD) and tPA-NADPH/*Rpep* (tPA-NADPH/R5Cs-sCR5) are utilized to enhance its clarity for readers.